# Role of Biochar in Improving Sandy Soil Water Retention and Resilience to Drought

**Ling Li [1],\*, Yong-Jiang Zhang [2] , Abigayl Novak [1] , Yingchao Yang [3] and Jinwu Wang [4]**

1  School of Forest Resources, University of Maine, Orono, ME 04469-5755, USA; abigayl.novak@maine.edu
2  School of Biology and Ecology, University of Maine, Orono, ME 04469-5751, USA; yongjiang.zhang@maine.edu
3  Department of Mechanical Engineering, College of Engineering, University of Maine, Orono, ME 04469-5711, USA; yingchao.yang@maine.edu
4  USDA Forest Service, Forest Products Laboratory, Madison, WI 53726, USA; jinwu.wang@usda.gov
\*  Correspondence: ling.li@maine.edu

**Abstract:** In recent years, plants in sandy soils have been impacted by increased climate variability due to weak water holding and temperature buffering capacities of the parent material. The projected impact spreads all over the world, including New England, USA. Many regions of the world may experience an increase in frequency and severity of drought, which can be attributed to an increased variability in precipitation and enhanced water loss due to warming. The overall benefits of biochar in environmental management have been extensively investigated. This review aims to discuss the water holding capacity of biochar from the points of view of fluid mechanics and propose several prioritized future research topics. To understand the impacts of biochar on sandy soils in-depth, sandy soil properties (surface area, pore size, water properties, and characteristics) and how biochar could improve the soil quality as well as plant growth, development, and yield are reviewed. Incorporating biochar into sandy soils could result in a net increase in the surface area, a stronger hydrophobicity at a lower temperature, and an increase in the micropores to maximize gap spaces. The capability of biochar in reducing fertilizer drainage through increasing water retention can improve crop productivity and reduce the nutrient leaching rate in agricultural practices. To advance research in biochar products and address the impacts of increasing climate variability, future research may focus on the role of biochar in enhancing soil water retention, plant water use efficiency, crop resistance to drought, and crop productivity.

**Keywords:** biochar; drought; porosity; sandy soil; water retention



## 1. Introduction

The projected increase in climate variability [1,2] poses an enormous threat to agricultural systems [3–6]. For agricultural systems with sandy soils, the impacts will be exacerbated due to their low water retention and capacity to buffer increasing variability in rainfall. Although certain types of crops can thrive and remain productive in sandy soils, such as wild (or lowbush) blueberries (*Vaccinium angustifolium Ait*.), these crops consume considerable amounts of water and fertilizer because the coarse texture of sandy soils has quick water drainage. Additionally, the sandy soils are easy to warm up in spring and tend to dry out in summer, and suffer from low nutrients that are washed away by rain. The severity, duration, and frequency of drought are predicted to increase in many regions of the world, including New England, USA [1,2,7].

Furthermore, the elevation of the annual average global temperature would lengthen the warm season (defined as when the average daily temperature is above freezing). For example, it is reported that the warm season in Maine, USA, has been extended by two weeks over the past 100 years and will continuously increase by approximately two more weeks over the next 50 years in Maine [8]. The extended warm season may result in

increased water use and enhanced drought effects on unirrigated farmland in dry seasons or excessive amounts of nitrate leaching from the root zone in wet seasons.

The current solutions for mitigation of drought effects on crop growth fall in two categories: (1) applying biochar as an addition in soils [9–17], and (2) growing cover crops to improve water infiltration and surface soil quality [18]. Here, we will review the biochar application as a possible technique for enhancing the capacity of agricultural systems to buffer increasing drought effects. The characteristics of biochar will be briefly reviewed, given that they vary greatly with different biomass feedstocks (e.g., manure, agricultural residues, forest residues, and wood processing residues, etc.) and thermal conversion pathways (such as temperature, residence time, particle size of gasification and pyrolysis). There is a strong controversy with respect to the effects of biochar as an addition to sandy soils. Several studies have shown positive effects of biochar on soil water retention [19–33] others failed to provide promising results [32,34,35]. Due to the inconsistent research findings, it is worth understanding the functions and mechanisms of biochar in modifying the water and nutrition retention of sandy soils from the points of view of fluid mechanics. Therefore, this review aims to (1) discuss the physicochemical properties of biochar that significantly influence the water retention of sandy soils, (2) provide post-treatment methods to modify the key physicochemical properties of biochar, (3) summarize the effective management practices in the literature regarding the application of biochar for enhancing the water and nutrient retention of sandy soils and crop growth, and (4) identify current knowledge gaps, research problems, and priorities.

## 2. Soil Characteristics and Moisture Contents

### 2.1. Soil Characteristics

According to the United States Department of Agriculture (USDA) soil classification, sandy soil contains less than 10% clay, less than 15% silt, and at least 85% sand, while loamy sand soil consists of 10% to 20% clay, less than 30% silt, and 70% to 85% of sand (bottom-left corner of the soil texture triangle in Figure 1). The sizes of sand in sand soils are classified into five (5) categories, including very fine sand (0.05–0.10 mm in diameter), fine sand (0.10–0.25 mm in diameter), medium sand (0.25–0.50 mm in diameter), coarse sand (0.50–1.00 mm in diameter), and very coarse sand (1.00–2.00 mm in diameter).

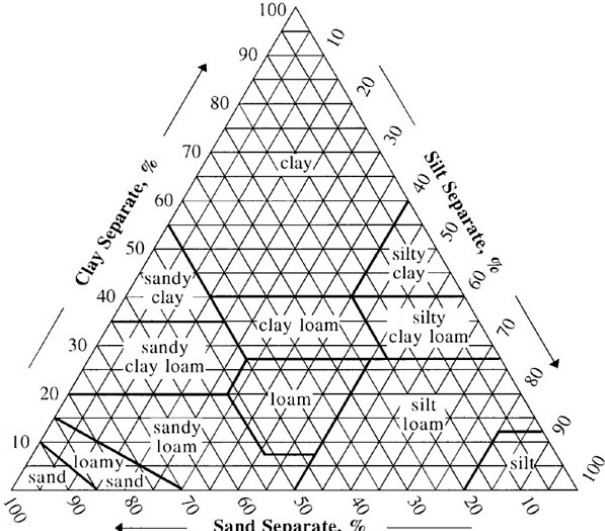

**Figure 1.** USDA soil textural triangle [36].

Based on the classifications of soil pore size by the Soil Science Society of America, macropores (>80 μm in diameter) contribute to the rapid flow of water (including nutrient-rich water) through soil by gravity, and mesopores (30 μm to 80 μm in diameter) allow water to move in response to matric potential differences (i.e., from wetter to drier areas).

It is micropores (5 μm to 30 μm in diameter) that hold water in place by capillarity to maximize water and nutrient retention in soils. The smallest pore sizes are ultra-micropores (0.1 μm to 5 μm in diameter) and cryptopores (<0.1 μm in diameter). The low water and nutrient retention capacity of sandy soil can be attributed to the relatively small surface area, porosity, and a large range of pore distribution of its soil particles [9]. Soil particle specific surface area (area $m^2/g$) determines nutrient retention. The number of cations that hold nutrient anions in the soil increases with the surface area. The surface areas of coarse and fine sands are 0.01 $m^2/g$ and 0.1 $m^2/g$, respectively. By contrast, the surface area of biochar is between 5 $m^2/g$ and 100 $m^2/g$, up to 1000 $m^2/g$ for activated biochar [37–40]. Adding biochar would bring a net increase in surface area of the sandy soil.

### 2.2. Soil Moisture Contents

The soil moisture ($\theta$) can be expressed gravimetrically ($\theta_g$, g water/g soil) or volumetrically ($\theta_v$, $cm^3$ water/$cm^3$ soil). To quantify soil water holding capacity, three key soil moisture parameters: (1) field capacity (FC), (2) plant permanent wilting point (PWP), and (3) plant available water content (AWC) can be evaluated and compared. FC ($\theta_{fc}$) is the maximum water content held in the soil after the drainage has stopped. PWP ($\theta_{pwp}$) is the minimum water content at the state when the plant dies. AWC ($\theta_{awc}$) is the difference between the field capacity and the permanent wilting point. The PWP is closely related to the specific surface area of soil, whereas the AWC does not depend on it [41]. All three parameters of sandy soil are much lower than those of other soil types (Figure 2). The sandy soil is very coarse and its $\theta_{awc}$ is close to zero due to the nearly same $\theta_{fc}$ and $\theta_{pwp}$. Increasing $\theta_{awc}$ means modifying the soil structure towards higher porosity with smaller pore sizes (shifting the soil structure to the left). Moreover, any ways to improve the field capacity and decrease the wilting point would increase the available water for plants.

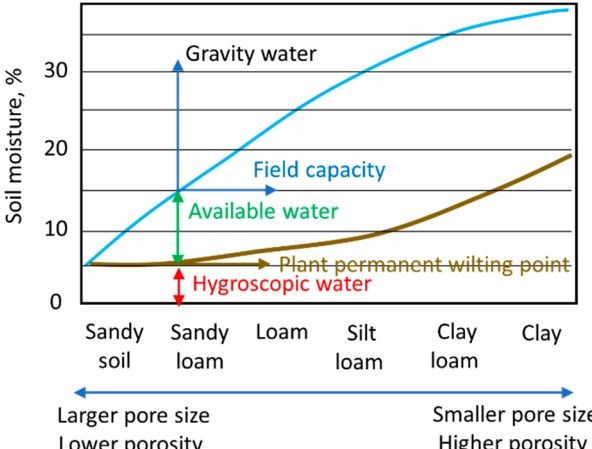

**Figure 2.** Soil moisture content for various soil textures including hydroscopic, available and gravity water. The blue line indicates the field capacity of a spectrum of soils with a range of pore sizes and other characteristics where excess water (gravity water) has drained. The brown line shows the minimum amount of water in various soils that a plant needs not to wilt (the plant permanent wilting point in the given soil).

The soil moisture parameters are influenced by gravity, the forces of capillarity (pressure), adsorption (electrostatic), and osmosis (solute). At the same time, the plant $\theta_{awc}$ is significantly impacted by the forces of capillarity and adsorption. The plant $\theta_{awc}$ can only realistically be determined from measurement of the water potential ($\varphi$) of the soil, after gravitational drainage (~0 kPa) and when $\varphi$ is in the range of −1000 to −1500 kPa, as shown in Figure 3. The permanent wilting point being where plant water availability terminates due to the high counter pressure required by roots to extract it from the soil

particle matrix, as determined by the soil matrix potential ($\varphi_m$). The addition of biochar may alter the particle matrix of soil to increase the $\theta_s$ and $\theta_{fc}$ (Solid line in Figure 3).

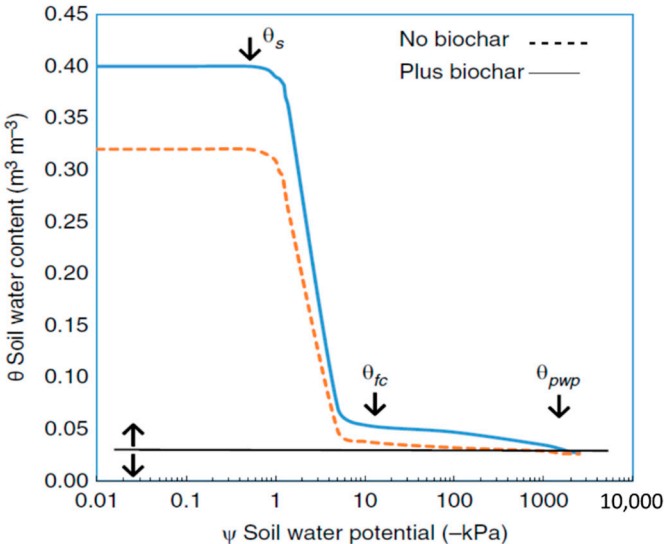

**Figure 3.** Relationship between soil water volumetric content ($\theta$) and soil water potential ($\varphi$) [42]. Reprinted from the reference with the permission from John Wiley and Sons, copyright 2018 British Society of Soil Science.

## 3. Characteristics of Biochar Related to Water and Nutrient Retention of Soil

### 3.1. Porosity of Biochar

Biochar derived from various biomass feedstocks has different pore sizes and pore distributions. In terms of formation of the pores, they can be grouped in two categories: residual macropores, being internal pores inherited from biomass feedstock structure; and pyrogenic nanopores, being internal pores produced at various thermal conversion conditions (such as temperature and residence time) [43]. The porosity of porous materials, including biochar, can be measured by different techniques, such as gas adsorption, mercury intrusion porosimetry, optical and electronic microscopy, etc.

Multiple biomass samples have been used to produce biochar. A detailed analysis of the volume of pore and pore distribution was performed on biochar samples derived from woody biomass (cedar (CE), cypress (CY), Moso bamboo (MB)), agricultural residues (rice husk (RH), sugarcane bagasse (SB)), poultry manure (PM), and wastewater sludge (WS) [44]. The biochar samples were produced through a slow pyrolysis process at three temperatures of 400 °C, 600 °C, 800 °C. The porosity, including total volume, pore sizes, and pore distribution of biochar samples, was measured by the mercury instruction porosimetry (MIP) method. A capillary-rise equation considering the capillary action of liquid water in small pores was employed to determine the size of capillary pores corresponding to AWC. The results showed that the optimal pore size for a capillary-rise of liquid water should range from 0.2 μm to 9 μm in diameter. This key information can be used to guide the selection of biochar materials. The total volume, volumes of different pore types, and volume of pores corresponding to AWC of the seven biochar materials have been comprehensively addressed in [44]. Sugarcane bagasse biochar has the greatest total volume of pores and volume of pores related to AWC, followed by wood-based biochar materials (CE, CY, and MB). The plot of AWC of biochar and volume of pores related to AWC of different biochar samples shows a good correlation between the volume of pores with a diameter in the range of 0.2 μm to 9 μm and AWC (Figure 4).

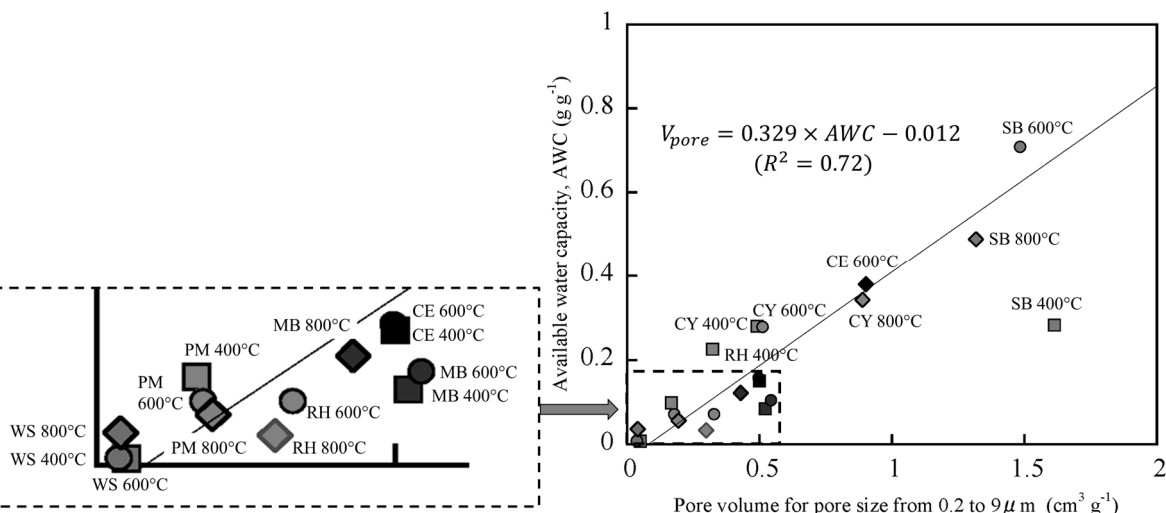

**Figure 4.** Plot of volumes of pore size from 0.2 μm to 9 μm and available water capacity (AWC) of biochar. AWC was calculated from water retention curves (WRCs) of biochar samples [44] reproduced from the reference, copyright 2019, CC-BY-4.0.

Other studies have also reported that biochar produced from cellulosic biomass such as grain husk [14], pig manure compost [45,46], cattle manure [47], cattle manure mixed with straw [47], chicken manure [47], chicken manure mixed with sawdust [47] and pig slurry (PC) [47], red oak (*Quercus rubra*) [15,22], ground wheat straw and wood pellets (69% Norway Spruce, 19% Beech plus other wood species) [25], North European grassland species [34], *Picea abies* (70%) and deciduous wood shavings of *Fagus sylvatica* (30%) [27], wheat and maize straws [33], mesquite feedstock/mesquite wood (*Prosopis* sp.) [32,48], and Pseudotsuga menziesii [24] have large portions of the volume of pores that are 1 (or smaller) to 10 μm in diameter, because the biochar inherited the architecture of the biomass [9,48–50]. When the biochar is added into the soils, such pores are important for improving AWC. Moreover, biochar particles might also interact with the soil and generate more interstitial pore space within the relevant size interval [48].

### 3.2. Hydrophobicity of Biochar

Hydrophobicity is another property of fresh biochar, which has a negative impact on its water-retention capability. Adsorption and condensation of moisture vapor on the surface of pores are reduced by the hydrophobicity of biochar, but the evaporation of moisture vapor is increased under dry conditions. The low surface energy on the hydrophobic surface of biochar allows a fast water-drop penetration throughout macropores (i.e., water drops run through the macropores quickly). The hydrophobicity of biochar is measured through a molarity-of-ethanol-droplet (MED) testing method, which is also called MED index [44,51,52]. The degrees of hydrophobicity can be classified into three categories: hydrophilic (<1 M (mol/L)), hydrophobic (1–2 M), strongly hydrophobic (2–3.5 M), or extremely hydrophobic (>3.5 M) [51].

It has been conclusively found in many studies [43,44,52] that biochar produced at relatively low temperatures (<400 °C) shows stronger hydrophobicity than the biochar produced at high temperatures (>500 °C), especially for wood-based biochar. The hydrophobicity of biochar can be explained by the presence of aliphatic compounds on the surface of biochar [23,52], which was verified by a strong correlation between hydrophobicity of biochar and the presence of alkyl functionalities in Fourier-transform infrared spectroscopy (FTIR) spectra ($R^2 = 0.87$, $p < 0.001$). The loss of labile aliphatic compounds in biochar samples derived from pig manure, crop residues, and municipal solid waste has been reported to cause the disappearance of hydrophobicity when the pyrolysis temperature is over 500 °C [53]. Omondi et al. showed in their review that for a variety of biochar feedstocks and soil textures, high temperature (>500 °C) amendment significantly

increases mean soil saturated hydraulic conductivity by 40%, while variation from 250 °C to 500 °C had no effect [31]. Feedstock type was also found to influence biochar hydrophobicity [34,38,53]. For example, biochar derived from corn stover and switchgrass had a significantly higher hydrophobicity than that derived from ponderosa pine wood.

In addition to changing the pyrolysis temperature, the hydrophobicity of biochar can also be modified by post-treatment. It was revealed that an acid oxidization treatment ($HNO_3$/$H_2SO_4$ mixed solution with a volume ratio of 1:3) could convert the hydrophobic biochar to hydrophilic biochar [39]. Other studies showed that aging of biochar in the soil could increase the hydrophilicity of biochar due to a surface oxidization process as biochar reacts with air and water in the soil to form carboxylate and other ionizable functional groups [22,54,55]. A novel method for the conversion of the hydrophobicity of carbonized wood is to spray Polydimethylsiloxane (PDMS) solution with a 1% concentration on the surface of biochar [56]. After the PDMS dries, a thin layer is coated on the surface of biochar. PDMS film allows water vapor molecules to pass through due to a solution-diffusion process where a pressure difference is a driving force.

## 4. Biochar–Soil Mix for Increasing Water and Nutrient Retention of Soil

### 4.1. Modification of Soil Porosity

The addition of biochar particles with different sizes and shapes in sandy soils can reduce the volume of large space between soil particles (i.e., interpore) and increase the portion of micropores (5 μm to 30 μm in diameter) contributed by the intrapores of biochar. The influence of biochar intrapores and biochar particle shape on the water retention of biochar–soil mixtures has been investigated [32], revealing that the sandy soil contains medium to coarse sand with particle size in the range of 0.25 to 0.85 mm. In [32], biochar particles were grouped into fine size (<0.25 mm), medium size (0.25 to 0.85 mm) and coarse size (0.85 to 2 mm). Two (2) w.t.% of fine, medium, and coarse biochar particles were mixed with sand, respectively. Then the soil moistures of $\theta_{fc}$, $\theta_{pwp}$, $\theta_{awc}$ of biochar-sand mixture samples were measured (Figure 5). Significant increases in $\theta_{fc}$ and $\theta_{awc}$ were observed in the two types of mixtures (medium biochar-sand, and coarse biochar-sand) while the wilting point only slightly increased. It was reported that biochar additions could convert drainable pores between soil particles (pore size within 60 μm to 300 μm in diameter) into water retaining pores (pore size within 0.2 μm to 60 μm in diameter) and, therefore, the plant $\theta_{awc}$ of sandy soil with additions of biochar was significantly increased [41,42]. The increase of $\theta_{awc}$ was affected by the amount of biochar addition and size of biochar particles.

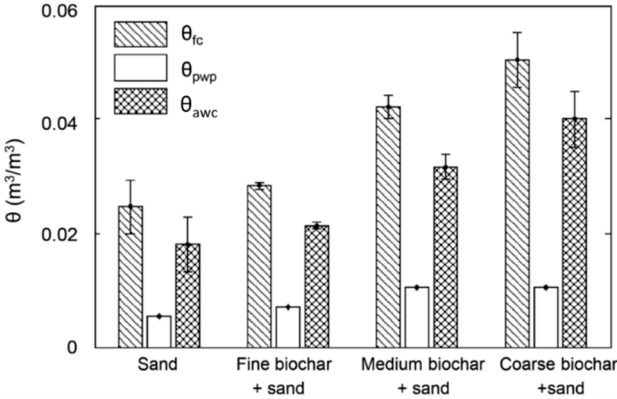

**Figure 5.** Field capacity ($\theta_{fc}$), permanent wilting point ($\theta_{pwp}$), and plant available water content ($\theta_{awc}$) of sand, and three biochar–soil mixtures [32]. Reprinted from the reference with the permission, © 2021 Liu et al., CC-BY-4.0.

## 4.2. Soil Microcosm Experiments and Leaching Tests

The performance of soils with biochar additions in terms of water retention is often evaluated by soil microcosm experiments and leaching tests. Basso et al. investigated the influence of the addition of biochar on the water-holding capacity of sandy loam soil, and the influence of the depth of biochar incorporation on the rate of biochar surface oxidation in the amended soils [22]. They mixed hardwood fast pyrolysis biochar with a soil (0%, 3%, and 6% w.t./w.t.) and placed the soil/biochar samples into columns in either the bottom 11.4 cm or the top 11.4 cm to simulate deep banding in rows (DBR) and uniform topsoil was mixing (UTM) applications, respectively. Four sets of 18 columns were incubated at 30 °C and 80% RH. Every 7 days, 150 mL of 0.001 M calcium chloride solution was added to the columns to produce leaching. Sets of columns were harvested after 1, 15, 29, and 91 days. The addition of biochar increased the gravity-drained water content by 23% relative to the control. Biochar did not affect the cation exchange capacity (CEC) of the soil in their study. Their results suggest that biochar added to sandy loam soil increases water-holding capacity and might increase water available for crop use.

## 4.3. Field Trials of Soil Mixed with Biochar

The common approach of biochar application is to blend biochar particles with fertilizers or composts together and spread the mixture in the field [57–59]. A wide range of biochar spreading rates has been used, varying from about 4 t/ha to 20 t/ha, depending on the soil texture. In [41], the percentage of biochar addition in sandy soil could be as high as 45% by volume (70 t/ha), which increases the available water content by 18%. Another study [33] reported an 8-year field trial with a goal of studying the effects of successive additions of high-dose maize-cob-derived biochar (9.0 t/ha per year, HB), low-dose maize-cob-derived biochar (4.5 t/ha per year, LB), straw return (SR) and control (no biochar or straw, CK) on hydraulic conductivity and water retention in the upper 10 cm of a sandy loam soil from the North China Plain. The results showed that the retention of soil water was improved under HB soil with evidence of plant-available water content increased by 17.8 (HB treatment) and 10.1% (LB treatment). Biochar used in the coarse-textured soil enhanced saturated hydraulic conductivity and water-retention capacity. Blue Leaf Inc. (Quebec City, QC, Canada) conducted a large-scale field trial for soybean crops. The commercial biochar blended with manure was spread in soybean fields at a rate of 3.9 t/ha, and a 19% yield increase was achieved [60].

## 5. Perspectives and Future Research Priorities

### 5.1. Long-Term Effectiveness of Biochar Added in Sandy Soil

A reasonable number of research studies have presented lab results providing evidence that biochar as a soil amendment could increase the moisture contents (field capacity and plant available water content) of coarse soils, such as sandy soil, and thereby improve the water and nutrition retention of the soils [19–30,34,42]. Some studies [34,35], however, found no significant effect of biochar application on the water retention in loamy sand soil and coarse sand soil in the field tests. This might be due to the fact that the severe hydrophobicity of biochar prevents water from infiltrating into the biochar [34]. Therefore, some post-treatment methods for changing the hydrophobicity of biochar may be needed (as discussed in this review). The effects of biochar on improving the water retention of soil need more long-term field tests to confirm. There is still limited study on how biochar influences the soil water retention in temperate soils and its effects on plant response to drought stress [42]. More studies are needed to investigate the relationship among biochar addition, soil water retention, plant water consumption, and water use efficiency. The long-term effects of biochar addition on soil and plant water relations also need to be studied.

*5.2. Available Sources of Biochar*

When applying biochar-based products in large-scale agricultural production systems, the feasibility, economic profitability, and sustainable supply availability of biochar should be considered. Fast pyrolysis and gasification are two processes to obtain biochar cost-effectively. However, as a byproduct or waste stream, modifications of the biochar are needed through post-treatment methods. Slow pyrolysis provides a possible process for producing so-called "engineered" biochar (e.g., activated carbon with an extremely large surface area) using pre-treated biomass feedstocks. Since the primary product of this process is biochar, the relatively high production cost might be a factor limiting the use of biochar in large-scale agricultural systems. A potential biochar source might be industrial combined heat and power (CHP) systems. Biochar waste, as a residual due to incomplete combustion in CHP systems, is currently disposed of in landfills. Using this biochar waste in agricultural production systems would reduce the negative impacts of waste disposal on environments and bring additional revenues to the owners of CHP. For instance, it can reduce the need and space for landfills, decrease costs of landfill disposal, including handling and transferring, waste tax, or deposit fee.

*5.3. New Nutrient-Rich Biochar Pellets*

Biochar, which is itself derived from biomass feedstocks through thermal conversion processes, only provides trace nutrients from the source materials but no other needed nutrients (such as nitrogen, phosphorous, and potassium) to plants/crops. In management practices, biochar can be applied together with fertilizer or compost in the fields to facilitate and enhance fertilizer use efficiency. The drawbacks of fine particulate biochar are significant weight loss of biochar during handling and transport to fields and potential chronic damage to the human respiration system [57–59]. Densified biochar pellets were then developed to alleviate the loss of fine biochar particles and reduce the costs of storage, transport, and handling [17,45,46,61,62]. Moreover, biochar pellets can function as a nutrient carrier to increase nutrient use efficiency by reducing the nutrient release rate. In addition, pelleting biochar may alter the pore size distribution of biochar particles by crushing the macropores inherited from biomass feedstock structure and, thereby, increase the portion of micropores that correspond to plant available water content.

## 6. Conclusions

Biochar has been investigated for four major applications: soil improvement (for improved productivity), waste management (reduced pollution), climate change mitigation (carbon storage), and energy production. The findings have shown that biochar has great potential as a soil amendment for socioeconomic and environmental benefits. This review focused on the influence of biochar incorporation in sandy soil on the water and nutrient (soluble) retention of soil, and its potential in enhancing the capacity of soils in buffering the increasing drought. The current findings show that the biochar effect on soil water retention is still under debate. The biochar with key physicochemical properties can alter the texture of sandy soil and soil moisture parameters and offer a mechanism of water storage. The hydrophobicity of biochar could be a major factor limiting its function in enhancing soil water retention. Therefore, several post-treatment methods to convert biochar from hydrophobic to hydrophilic were summarized. It is expected that this review could draw significant attention from the biochar community to put effort into processing, characterization, application of biochar as well as revealing the interdependence between them in order to obtain convincing findings.

**Author Contributions:** Conceptualization, L.L.; methodology, L.L.; writing—original draft preparation, L.L.; writing—review and editing, Y.-J.Z., A.N., Y.Y. and J.W.; visualization, A.N.; supervision, L.L.; project administration, L.L.; funding acquisition, L.L., Y.-J.Z. and Y.Y. All authors have read and agreed to the published version of the manuscript.

**Funding:** This project was supported by the USDA National Institute of Food and Agriculture (NIFA), McIntire-Stennis Project Number ME042002 through the Maine Agricultural & Forest Experiment Station. Maine Agricultural and Forest Experiment Station Publication Number 3794; USDA NIFA Hatch Fund through the Maine Agricultural Forest Experiment Station (ME022021) 2020 Maine Agricultural Development Grant; the U.S. Department of Agriculture's Agricultural Research Service (USDA ARS Agreement No. 58-0204-6-003 & No. 58-0204-9-166); the U.S. Department of Interior, Bureau of Reclamation under Grant R19AC00116; the Wild Blueberry Commission of Maine, Maine Department of Agriculture, Conservation and Forestry (ADG and SCBG).

**Institutional Review Board Statement:** Not applicable.

**Informed Consent Statement:** Not applicable.

**Acknowledgments:** The authors would like to thank Yong-Jiang Zhang's graduate student Rafa Tasnim for discussing the experience in biochar as a soil amendment in wild blueberry fields with the authors.

**Conflicts of Interest:** The authors declare no conflict of interest.

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
