# Peer review of "Role of Biochar in Improving Sandy Soil Water Retention and Resilience to Drought"

_water, doi:10.3390/w13040407_

Round 1

Reviewer 1 Report

This paper illustrated the role of biochar in enhancing water retention. The existing physicochemical properties of biochar were fully summarized. Besides, the performance of biochar on the plant growth, development and yield were reviewed. The capability of biochar in reducing fertilizer drainage through increasing water retention can improve productivity and reduce the nutrient leaching rate in agricultural practices. However, I have a few concerns for the processing of this manuscript:

  1. Please emphasize the novelty of this review in the Abstract, an acceptable review should not only summarize the information but put forward practical solutions, new findings and future perspectives.
  2. Please improve the language and check grammar mistakes carefully.
  3. In introduction, please provides a brief summary on the advanced technologies or current used technologies for in Improving Sandy Soil Water 2 Retention and Resilience to Drought
  4. Please add a subsection on future outlook and the limitations of this review
  5. Conclusions changed to 6 Conclusion

Author Response

Reviewer 1:

1) In the abstract, one sentence was added to emphasize the novelty of this review. See lines 18-20.
2)  Language and grammar check was completed by one co-author who uses English as mother language.
3)  A brief summary of techniques for drought mitigation was added in the introduction section. See lines 49-51.
4) We already addressed the future outlook in section 5. Therefore, we kindly decline this suggestion.
5) This grammar error was revised.

Reviewer 2 Report

Following an overall inquiry into the reviewed article, I consider it to be a very interesting investigation of Role of biochar in improving sandy soil water retention and resilience to drought.

The manuscript contains interesting and valuable data, which have been correctly interpreted. Organization and clarity of the manuscript is also generally good. The paper resolves an elaborate multidisciplinary topic and meets formal layout standards and default criteria, imposed on such articles.

Thereby, I recommend its issuance.

I would like to ask the authors to marginally mention the following article in the introduction:

Viglašová E. etl al.:

- Production, characterization and adsorption studies of bamboo-based biochar/montmorillonite composite for nitrate. Waste Management 79: 385-394 (2018).

- Engineered biochar as a tool for nitrogen pollutants removal: preparation, characterization and sorption study. Desalin. Wat. Treat. 191: 318-331 (2020).

Author Response

Reviewer 2:

The two reference articles recommended by this reviewer talked about the application of biochar in wastewater treatment and they are very good references. However, this subject is marginally related to the subject of this review, the two references will be used in another work that is relevant to wastewater treatment.